## RESEARCH ARTICLE

# Genetic variation in the FMO and GSTO gene clusters impacts arsenic metabolism in humans

Lizeth I. Tamayo[1], Lin Tong[1], Tetiana Davydiuk[2], Donald Vander Griend[3], Syed Emdadul Haque[4], Tariqul Islam[4], Farzana Jasmine[5], Muhammad G. Kibriya[5], Joseph Graziano[6], Lin Chen[1], X. Chris Le[2], Habibul Ahsan[1,5,7,8,9], Mary V. Gamble[6], Brandon L. Pierce[1,7,8]*

1 Department of Public Health Sciences, University of Chicago, Chicago, Illinois, United States of America, 2 Department of Laboratory Medicine and Pathology, University of Alberta, Edmonton, Canada, 3 Department of Pathology, University of Illinois Chicago, Chicago, Illinois, United States of America, 4 University of Chicago Research Bangladesh, Dhaka, Bangladesh, 5 Institute for Population and Precision Health (IPPH), Biological Sciences Division, University of Chicago, Chicago, Illinois, United States of America, 6 Department of Environmental Health Sciences, Columbia University, New York, New York, United States of America, 7 Department of Human Genetics, University of Chicago, Chicago, Illinois, United States of America, 8 Comprehensive Cancer Center, University of Chicago, Chicago, Illinois, United States of America, 9 Department of Medicine, University of Chicago, Chicago, Illinois, United States of America

* brandonpierce@uchicago.edu

## Abstract

### Background

In Bangladesh, > 50 million individuals are chronically exposed to inorganic arsenic (iAs) through drinking water, increasing risk for cancer and other iAs-related diseases. Previous studies show that individuals' ability to metabolize and eliminate iAs, and their risk of toxicity, is influenced by genetic variation in the *AS3MT* and *FTCD* gene regions.

### Methods

To identify additional loci influencing arsenic metabolism, we used data from Bangladeshi individuals to conduct genome-wide association analyses of the relative abundances of arsenic species measured in both urine (n = 6,540) and blood (n = 976). These species include iAs, monomethylated arsenic (MMA) and dimethylated arsenic (DMA) species.

### Results

In analyses of urine arsenic species, we identified a novel association signal in the FMO gene cluster (1q24.3), with the lead SNP residing in *FMO3* (MMA% P = 4.2x10⁻¹⁶). In analyses of blood arsenic species, we identified an additional signal in the FMO cluster, with the lead SNP residing in *FMO4* (DMA% P = 2.3x10⁻²²) and a novel signal at 10q25.1, with the lead SNP in *GSTO1* (DMA% P = 5.3x10⁻¹³). Lead SNPs at *FMO3* and

**Data availability statement:** GWAS summary statistics are provided at (https://uchicago.box.com/s/yfzeuvr9xd2n8k5yxbq666qqzhone-sa1). Individual-level data are available upon reasonable request to the IPPH research group at IPPH@uchicago.edu.

**Funding:** This study was funded by NIEHS R35 ES028379 (to B.L.P.), NIEHS P30 ES027792 (to H.A.), NIDDK R01 DK123285 (to M.V.G.), and NIEHS R21 ES035491 (to D.V.G. and B.L.P.). The Funders have no role in study design, data collection and analysis.

**Competing interests:** The authors have declared that no competing interests exist.

*GSTO1* are associated with the splicing of *FMO3* and *GSTO1*, respectively, in multiple tissue types, but also contain missense variants. The lead SNPs at *FMO4* are associated with *FMO4* expression level in multiple tissue types. These newly identified SNPs did not show a clear association with risk for arsenic-induced skin lesions ($P > 0.05$), based on 3,448 cases and 5,207 controls.

## Conclusion

We identified novel loci influencing arsenic metabolites measured in both urine and blood. FMOs are involved in the oxidation of xenobiotics but have no known direct role in arsenic metabolism, while GSTO1 has a well-established role in catalyzing the reduction of arsenic species. The novel associations we report appear specific to blood or urine, with no detectable impact on skin toxicity risk, pointing to complexities in arsenic metabolism and its genetic contributors that require further study.

### Author summary

We conducted a study to better understand how humans differ in their ability to process arsenic, a harmful chemical found in drinking water that affects millions globally, particularly in Bangladesh. While we've already identified variants in specific genes that impact arsenic metabolism and elimination (*AS3MT* and *FTCD*), our research aimed to discover additional genetic influences. Using genetic information and arsenic measurements from both urine and blood samples of arsenic-exposed individuals, we identified new genetic regions, specifically within the FMO and GSTO gene clusters, that impact arsenic metabolism. Interestingly, our findings show that some genetic variants affect arsenic metabolites measured in urine, others in blood, and some in both. Unlike previously identified genetic factors that are linked to arsenic-induced skin lesions, these newly discovered genetic influences do not appear to have a clear association with skin toxicity. This finding points to complexities in arsenic metabolism we do not yet understand, with different genes affecting its processing in distinct ways depending on where arsenic species are measured in the body. Our work expands knowledge of how genetic variation influences arsenic metabolism and could help inform future public health strategies to protect vulnerable populations.

### Introduction

Inorganic arsenic (iAs) is a human carcinogen [1], and exposure to iAs affects >200 million individuals worldwide through drinking water [2,3]. In Bangladesh, > 50 million individuals are chronically exposed to iAs through drinking water from naturally-contaminated wells that have some of the highest iAs concentrations reported in the world [3,4]. Chronic exposure to iAs above the World Health Organization (WHO) safety standard (>10 µg/L) increases the risk for multiple diseases, including cancer

[5–7]. The most common signs of toxicity caused by iAs exposure are skin lesions, which appear as hyperpigmentation during early exposure and keratosis at advanced stages of exposure [8].

The liver is believed to be the primary site for the metabolism of ingested iAs, where iAs can undergo methylation catalyzed by arsenite (+3) methyltransferase (*AS3MT*) [9–11]. While the sequence of reactions involved in iAs biotransformation is still under debate [12], the resulting methylated arsenicals are well defined, and these include monomethylated forms of arsenic (MMAs) and dimethylated forms of arsenic (DMAs) [13]. DMA has a shorter half-life in circulation and is more readily eliminated in urine compared to MMA and iAs [14]. Consequently, DMA constitutes the majority of excreted arsenic species [15,16]. Hence, an individual's arsenic metabolism efficiency (AME) can be defined as their capacity to methylate arsenic and generate DMA. The percentage of DMA among the total arsenic species in urine (DMA%) is commonly used as an indicator of AME [17–19].

There is inter-individual variation in AME [20–24] and this variability impacts the internal dose of arsenic and toxicity risk [4,25]. While age, sex, diet, body mass index, and smoking status are all likely to contribute to variability in AME [26], inherited genetic factors also play an important role. Prior GWAS and targeted sequencing studies of Bangladeshi individuals have identified four independent association signals for AME: three in the 10q24.32/*AS3MT* region (represented by rs4919690, rs11191492, and rs191177668) [20,27] and one in the *FTCD* gene (rs61735836) [19], a gene involved in histidine catabolism.

In addition to *AS3MT* and *FTCD*, there are likely to be other regions contributing to arsenic metabolism and toxicity risk [28]. Given the significant global health impact of arsenic exposure and the variability in AME among individuals, our objective is to identify novel genetic determinants of arsenic metabolism and toxicity phenotypes. Identifying such genetic determinants will implicate novel genes in AME, improving our understanding of biological processes underlying arsenic metabolism. Furthermore, knowledge of AME-associated variants could ultimately inform precision public health strategies focused on identifying highly susceptible groups who may benefit from targeted interventions, such as nutritional supplementation or enhanced exposure monitoring [29,30]. Here, we report the largest genome wide association study (GWAS) of urine arsenic species conducted to date and first GWAS blood arsenic species to identify inherited genetic effects on arsenic metabolism and toxicity.

## Methods

### Ethics statement

This research was approved by the Institutional Review Boards of the University of Chicago, Columbia University, and the Bangladesh Medical Research Council. Informed verbal consent was obtained from all participants.

### Inclusivity in global research

Additional information regarding the ethical, cultural, and scientific considerations specific to inclusivity in global research is included in the Supporting Information (S1 Checklist).

### Study participants

DNA samples were obtained at baseline recruitment from The Health Effects of Arsenic Longitudinal Study (HEALS) [31] and the Bangladesh Vitamin E and Selenium Trial (BEST) [32]. The HEALS cohort is prospective study of health outcomes associated with arsenic exposure through drinking water of adults in located in Araihazar, Bangladesh, a rural area with substantial exposure to arsenic through naturally contaminated drinking water from local wells. HEALS began in 2000 and is comprised of >20,000 adult participants (followed over 15–20 years to ascertain health outcomes), of which 6,540 have genome-wide SNP data. Trained study physicians conducted in-person interviews, clinical evaluations, and urine collection at baseline follow-up visits. BEST is a randomized chemoprevention trial evaluating effects of vitamin E

and selenium supplementation on skin cancer risk among arsenic-exposed individuals. BEST participants are residents of Araihazar, Matlab, and surrounding areas in Bangladesh. BEST has 7,000 adult participants, all with skin lesions at baseline, and 1,990 have existing genotype data. BEST uses many of the same study protocols as HEALS, including arsenic exposure assessment and biospecimen collection.

Some HEALS participants also participated in additional studies investigating correlates of AME and folate interventions to increase AME. We used data from 1,099 genotyped HEALS participants who participated in one of three such studies: the Nutritional Influences of Arsenic Toxicity study (NIAT, n = 163), the Folic Acid and Creatine Trial (FACT, n = 595), or the Folate and Oxidative Stress study (FOX, n = 341). These studies measured participants' arsenic species in both blood and urine. Data on blood arsenic species were available for 977 of the 1099 genotyped individuals for the NIAT (n = 110), FOX (n = 273) and FACT (n = 594) studies. Blood was collected at baseline (Week 0) for FOX, at Week 0 and 12 for NIAT, and at Weeks 0, 1, 12 and 24 for FACT. Urine was collected at Week 0 for FOX, weeks 0 and 12 for NIAT, and Weeks 0, 1, 6, 12, 13, 18, and 24 for FACT.

## Measurement of arsenic species

Both urinary and blood arsenic species were distinguished using a method described by Reuter et al [33]. This method entails using high-performance liquid chromatography separation of arsenobetaine, arsenocholine, $As^V$, $As^{III}$, MMA, and DMA followed by detection by inductively coupled plasma-mass spectrometry with dynamic reaction cell. Because $As^{III}$ can oxidize to $As^V$ during sample transport, storage, and preparation, we sum these two species to obtain total iAs (i.e., $As^{III}$ + $As^V$). The percentages of iAs, MMA and DMA in total arsenic were calculated by dividing the concentration of each species by the sum of iAs, MMA, and DMA. Data on arsenobetaine and arsenocholine, nontoxic organic arsenic from dietary sources, were not analyzed. Participants missing one of the four arsenic species were dropped from the analysis. **Generation of all urine arsenic species for HEALS and HEALS ancillary studies** (FOX [14], NIAT [34], FACT [35]) **were conducted at Columbia University's trace elements lab, as described** previously [36] with the exception of ~1,800 HEALS urine samples which were more recently analyzed for arsenic species at the University of Alberta in the laboratory of Dr. Chris Le. Both labs use HPLC-ICP-MS to generate urine arsenic species measurements. Generation of all data on blood arsenic species used in this work has been described previously (for FOX[14], NIAT[32], FACT[33]). Bio-specimen measurements were averaged when data from multiple time points was available, as this increased power to detect known signals *(AS3MT, FTCD)*.

## Genotype data

6,665 HEALS and 1,990 BEST participants have been genotyped using either Illumina's HumanCytoSNP-12 (299,140 SNPs), Infinium Multi-ethnic EUR/EAS/SAS arrays (1,475,140 SNPs), or the Global Screening Array (654,027 SNPs). Most participants with data from the HumanCytoSNP-12 also have complementary data from Illumina's exome array. For each array, we removed non-rs SNPs, SNPs with a call rate of <90%, monomorphic SNPs, and samples with a call rate <90%. The Michigan Imputation Server [37] was used to genotype unmeasured SNPs using the Haplotype Reference Consortium (HRC) reference panel.[31] Only high-quality imputed bi-allelic SNPs (imputation r2 > 0.3) and SNPs with minor allele frequency (MAF) >0.005 were retained (8,711,421 SNPs).

## Ascertainment of skin lesion status

HEALS and BEST participants were clinically assessed for skin lesions across the entire body at each study visit [31,32]. A protocol similar to the quantitative assessment of the extent of a body surface involvement in burn patients [38] was used to quantify the size, shape and extent of skin lesion involvement. The principle is based on dividing the entire body skin surface into 11 segments (e.g., front of arm, back of arm, face) and assigning percentages to each of them based on their size relative to the whole body [39]. There are 1,458 (prevalent & incident) skin lesion cases

among genotyped HEALS participants, 1990 genotyped cases from BEST, and 5,207 genotyped HEALS controls (with no history of lesions).

## Statistical analysis

GWAS of arsenic species percentages (iAs%, MMA%, and DMA%) were conducted using mixed linear models as implemented in the GCTA software [40] to account for relatedness among individuals in our sample (previously described [20]). All models are adjusted for age, sex, and genotype batch.

GWAS of skin lesions status was run separately by genotyping batch. There were 4,806 participants that were genotyped using the HumanCytoSNP-12 array (2395 cases, 2411 controls), 466 genotyped using the HumanCytoSNP-12 array without exome data (92 cases, 374 controls), 2,486 genotyped using the Multi-Ethnic-Array (913 cases,1573 controls) and 1133 genotyped using the global screening array (48 cases, 849 controls). The 5 GWAS were then meta-analyzed using PLINK 1.9.

We identified SNPs that pass a standard genome-wide threshold (P-value $< 5 \times 10^{-8}$). For signals identified, we (1) ensured all QC metrics for identified SNPs were acceptable, (2) searched for secondary signals after adjusting for the top SNP, and (3) examined linkage disequilibrium for the top SNPs.

To investigate the role of identified SNPs in gene regulation we leveraged data on expression quantitative trait loci (eQTL) and splicing QTLs (sQTL) from the Genotype-Tissue Expression (GTEx) project [41]. To determine whether a common causal variant was responsible for both GWAS and eQTL/sQTL signals at a given locus we used colocalization methods. We used the coloc R package version 5.2.2. [42] a Bayesian framework to assess the probability of shared causal variants at a given locus for multiple traits using SNP-based summary statistics (from GWAS or QTL studies). Coloc estimates the posterior probabilities of different colocalization hypotheses, including one shared causal variant (H4) and two distinct causal variants (H3). We used coloc's default priors.

## Results

### GWAS of arsenic species measured in urine

A GWAS of arsenic species measured in urine (DMA%, MMA%, and iAs%, distributions shown in S1 Fig) among 6,540 individuals identified previously reported associations in the *AS3MT* (DMA% P=$2.4 \times 10^{-51}$) and *FTCD* (DMA% P=$2.2 \times 10^{-48}$) regions (Fig 1). A novel association signal was identified in the FMO (Flavin-containing monooxygenase) gene cluster at 1q24.3, a region containing *FMO1, FMO2, FMO3*, and *FMO4*. FMOs metabolize xenobiotic chemicals through oxygenation, but currently there is no known role for FMO genes in arsenic metabolism. The minor allele (A, MAF=6.1%) at lead SNP rs12406572, located in intron 7 of *FMO3*, was associated with increased urine DMA% (beta=1.62, P=$3.9 \times 10^{-8}$) and decreased urine MMA% (beta=-1.3, P=$3.1 \times 10^{-16}$), but it did not show clear association with urine iAs% (beta=-0.28, P=0.22) (Fig 2). There was evidence of a suggestive secondary association signal in this region, particularly for MMA%, represented by lead SNP rs3754494 (beta=0.40, P=$6.5 \times 10^{-7}$) (S2 Fig). All associations were robust to adjustment for BMI and smoking status.

Among the SNPs showing the strongest evidence of association was rs12406572 (MMA% P=$3.2 \times 10^{-16}$), a missense variant in exon 7 of *FMO3* that codes for a Glu (E) to Gly (G) amino acid substitution (CADD score of 24.4; SIFT: 0.01/deleterious, PolyPhen: 0.86/possibly damaging).

Lead uMMA% SNP rs12406572 was associated with alternative splicing of *FMO3* in >10 GTEx tissue types, and the urine MMA% signal showed strong evidence of co-localization with a *FMO3* sQTL in at least 7 GTEx tissue types (S1 Table), including liver (PP4=0.98, Fig 3), adipose (subcutaneous) (PP4=0.98) and lung (PP4=0.98). In liver, co-localization was observed for five different intron excision events (S1 Table). For example, the MMA%-decreasing allele (A) was associated with increased excision of an alternative intron (chr1:171092790:171107675) that includes exon 3,

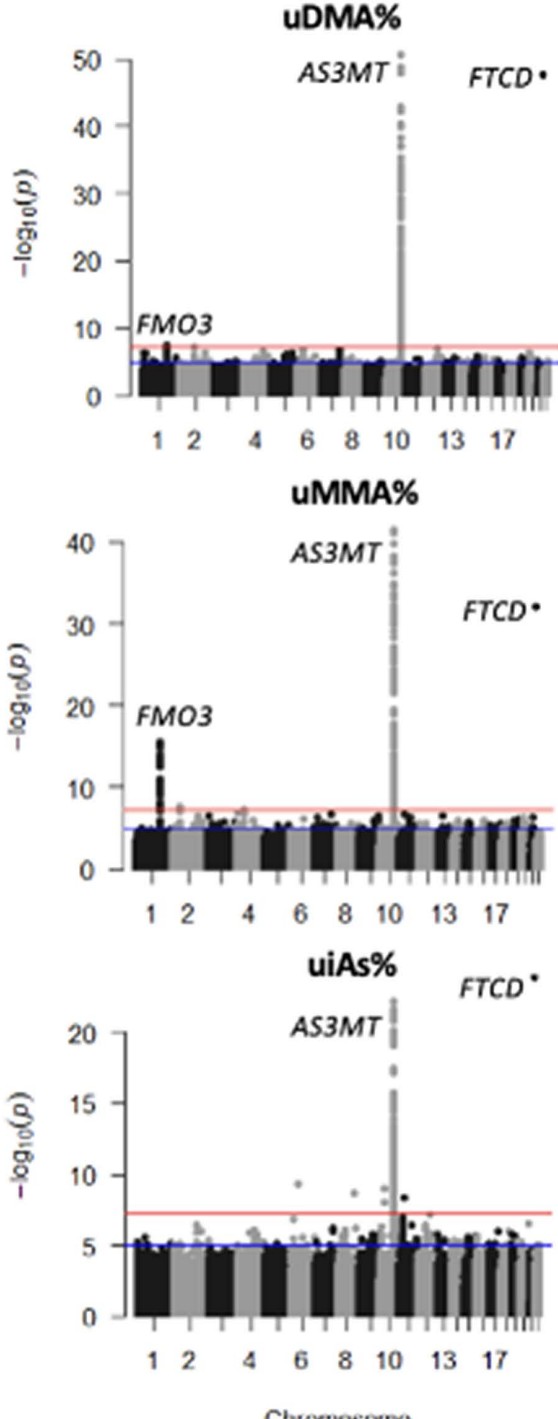

**Fig 1. GWAS of urine arsenic species percentages identifies known association signals at *AS3MT* and *FTCD* and a novel signal at 1q24.3/*FMO3*.**

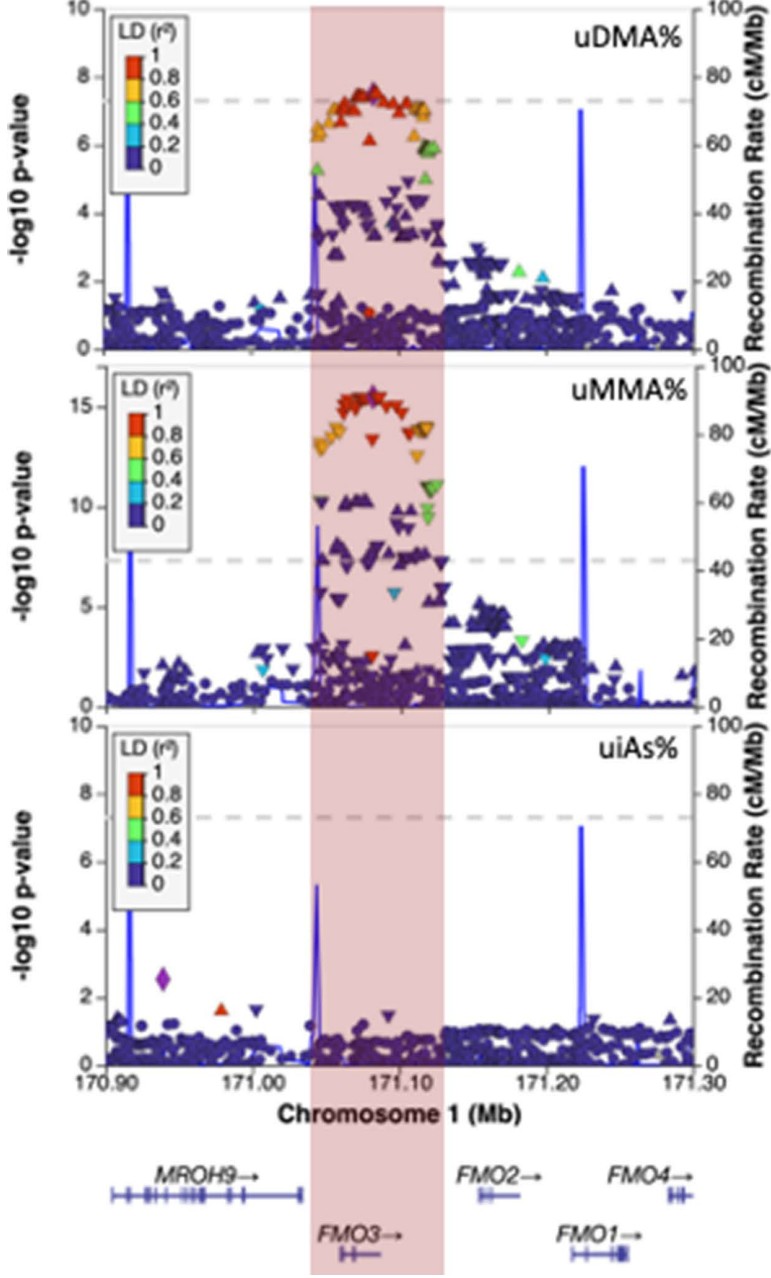

**Fig 2. The association signal for uDMA% and uMMA% at 1q24.3 spans the *FMO3* gene.**

leading to loss of exon 3 in the mature mRNA. These findings suggest that the alternative splicing of *FMO3* could play a role in modulating the metabolism of arsenic.

In HEALS, the MAF of lead SNP is 6%, consistent with the MAF observed in 1 KG BEB group (6%). However, the MAF rs12406572 varies substantially across 1 KG populations (3% in SAS, 12% in AMR, 17% in EAS, 17% in EUR, and 36% in AFR), suggesting that the causal variant at this locus may make a larger contribution to variability in arsenic methylation capacity in populations outside of South Asia.

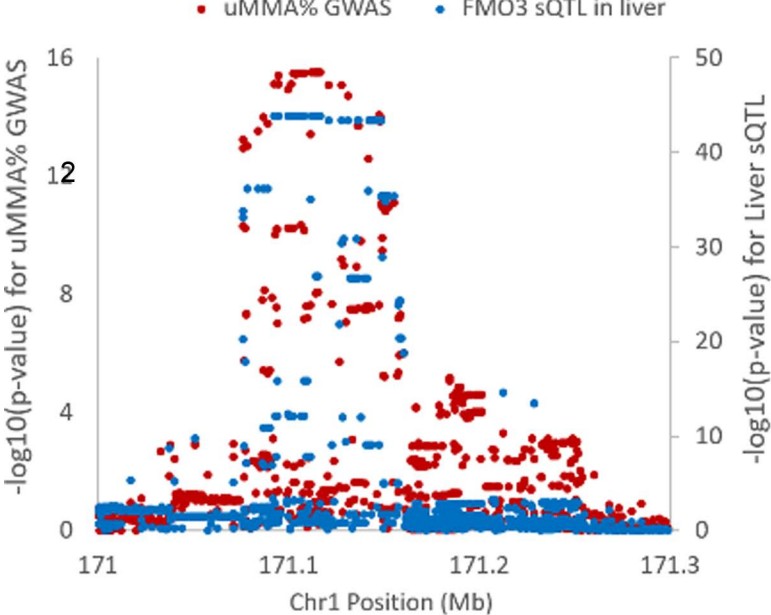

**Fig 3. Co-localization of the FMO3 association signal for urine MMA% with a sQTL for *FMO3* in liver.** The splicing phenotype shown is excision of intron chr1:171092790:171107675.

### GWAS of arsenic species measured in blood

A GWAS of blood arsenic species (bDMA%, bMMA%, and biAs%) among 976 individuals identified association signals in the *AS3MT* and *FTCD* regions (Fig 4) that appear very similar to the association signals observed for urine arsenic species in those regions, in terms of both lead SNPs and direction of association (S3 Fig). In addition, we observed two novel association signals, one in the FMO gene cluster (1q24.3), spanning the *FMO4* gene, and one signal spanning the *GSTO1* and *GSTO2* genes at 10q25.1, in close proximity (<2 Mb) to, but distinct from, the association signal at AS3MT/10q24.32 (Fig 5). All associations were robust to adjustment for BMI and smoking status.

The 1q24.3/FMO signal for blood DMA% spans the *FMO4* gene and is distinct from the association signal observed for urine arsenic species that spans *FMO3*. The minor allele (C, MAF = 45%) at uDMA% lead SNP rs10912834 was associated with decreased blood DMA% (beta = -2.13; P = 2.3x10$^{-22}$), increased blood MMA% (beta = 1.35; P = 1.2x10$^{-11}$), and increased blood iAs% (beta = 0.78; P = 1.7x10$^{-6}$). However, iAs% has a different lead SNP rs2011345 with minor allele C (MAF = 38%) associated with decreased blood iAs% (beta = -1.02; P = 3.4x10$^{-9}$). These two SNPs have an LD r$^2$ of 0.36, suggesting iAs% may have unique genetic determinants at this locus. The signal at *FMO4* observed for blood arsenic species was not observed for arsenic species in urine, and the signal at *FMO3* observed for urine metabolites was not observed for blood metabolites (Fig 6). The MAF of lead SNP rs10912834 in HEALS (45%) was consistent with the MAF observed in 1 KG BEB (46%) and SAS groups (45%). However, the MAF for rs10912834 varies somewhat across 1 KG populations (30% in AMR, 41% in EAS, 35% in EUR, and 57% in AFR). There was also evidence of a suggestive secondary association signal in this region, particularly for bDMA%, represented by lead SNP rs10798297 (beta = -1.16, P = 2.8x10$^{-5}$, S4 Fig.

The *FMO4* association signal for blood DMA% showed strong evidence of co-localization with a cis-eQTL for *FMO4* that was present in at least 10 GTEx tissue types, including liver (PP4 = 0.84), pancreas (PP4 = 0.98) and artery-tibial (PP4 = 0.86) (Figs 7 and S2). The minor, DMA%-decreasing allele (C) was associated with increased expression of *FMO4* in all of the GTEx tissues examined.

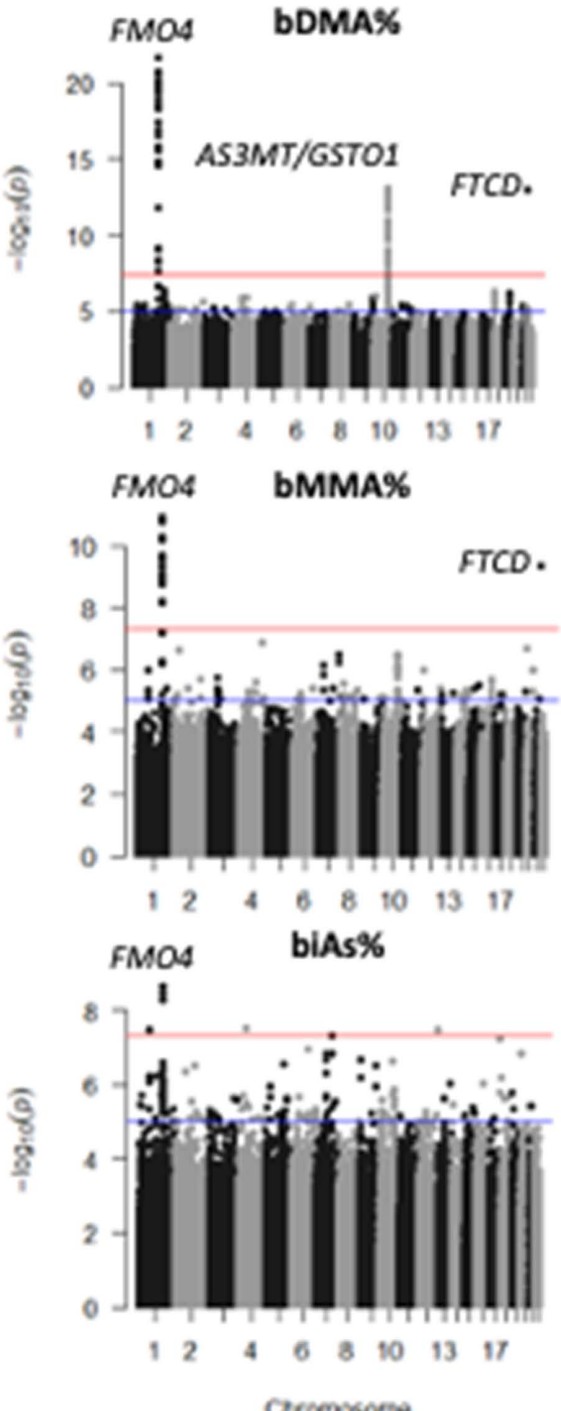

**Fig 4. GWAS of blood arsenic species percentages identifies association signals at 1q24.3 (*FMO4*), 10q24.32 (*AS3MT*), 10q25.1 (*GSTO1*), and 21q22.3 (*FTCD*).**

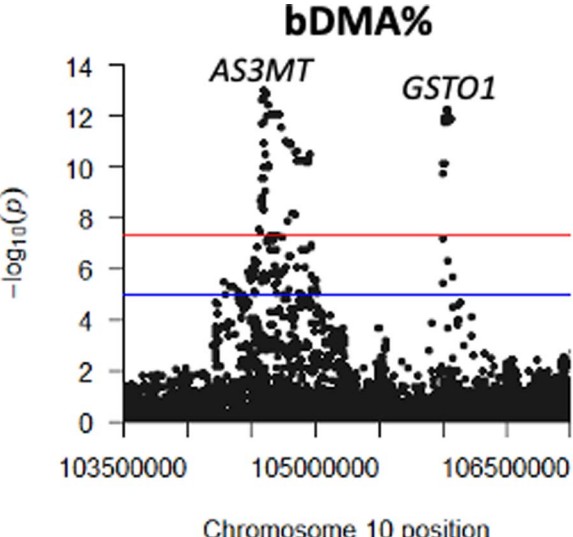

**Fig 5. Distinct association signals for blood DMA% at the *AS3MT* and *GSTO1* gene regions on chromosome 10.**

The association signal for blood arsenic species observed at 10q25.1 spanned *GSTO1* and *GSTO2*, with *GSTO1* having a well-established role in catalyzing the reduction of arsenic species (iAs$^V$ to iAs$^{III}$, MMA$^V$ to MMA$^{III}$, and DMA$^V$ to DMA$^{III}$) [43,44] The minor allele (T, MAF = 10.3%) at lead SNP rs34521730 was associated decreased blood DMA% (beta = -2.70; P = 5.3x10$^{-13}$), increased blood MMA% (beta = 1.72; P = 3.5x10$^{-7}$), and increased blood iAs% (beta = 0.98, P = 0.0004) (Fig 8). All associations were robust to adjustment for BMI and smoking status. SNPs in this region did not show evidence of association with arsenic species measured in urine.

Among the SNPs in the *GSTO1* region showing the strongest evidence of association was rs11509438 (DMA% P = 5.3x10$^{-13}$), a missense variant that codes for a GLU (E) to Lys (K) amino acid substitution. This variant has a CADD score of 0.46, a SIFT score of 0.36/tolerated, and a PolyPhen score of 0.007/benign.

Lead bDMA% SNP rs34521730 was associated with alternative splicing of *GSTO1* in >40 GTEx tissue types. We tested the bDMA% signal for colocalization with the corresponding sQTL in liver and observed strong evidence of colocalization (PP4 = 0.99, Fig 9). In liver (and in many other tissue types), the minor, DMA%-decreasing allele (T) at rs34521730 was associated with increased excision of intron chr10:104259798:104262979 (intron 3) and decreased excision of intron chr10:104263077:104266084 (intron 4) (S3 Fig). (intron numbers as based on canonical *GSTO1* isoform NM_004832.3/ ENST00000369713.9).

In HEALS, the MAF of lead SNP rs34521730 is 10%, which is fairly consistent with the MAF observed in 1 KG BEB (15%) and SAS groups (12%). The minor allele is less common across other 1 KG populations (2% in AMR, 1% in EAS, 3% in EUR, and 2% in AFR).

We also conducted a GWAS of blood total arsenic (computed as the sum of arsenic metabolites iAs$^{III}$, iAs$^V$, MMA, and DMA), but no clear associations were observed.

To assess the robustness of our findings, we also conducted a sensitivity analysis by adjusting our primary models for the top FMO3 (rs12406572, associated with uAsMet) and FMO4 (rs10912834, associated with bAsMet) SNPs for BMI and smoking status. The adjusted results remained consistent with the original analysis, indicating that the associations of interest are largely independent of these covariates beyond the arsenic methylation efficiency variables.

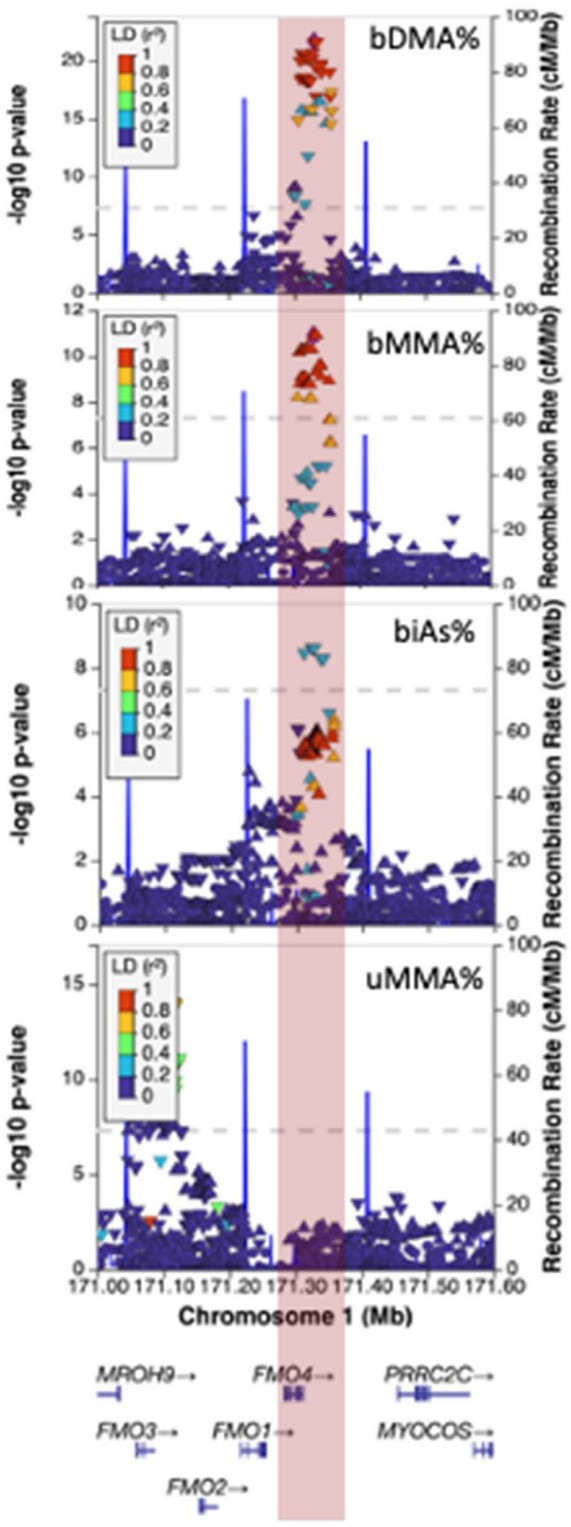

**Fig 6. Association of SNPs in the *FMO4* region with arsenic species measured in blood (bDMA%, bMMA%, and biAs%) and urine MMA% (bottom panel), LD is shown in relation to bDMA%/bMMA% lead SNP rs10912834.**

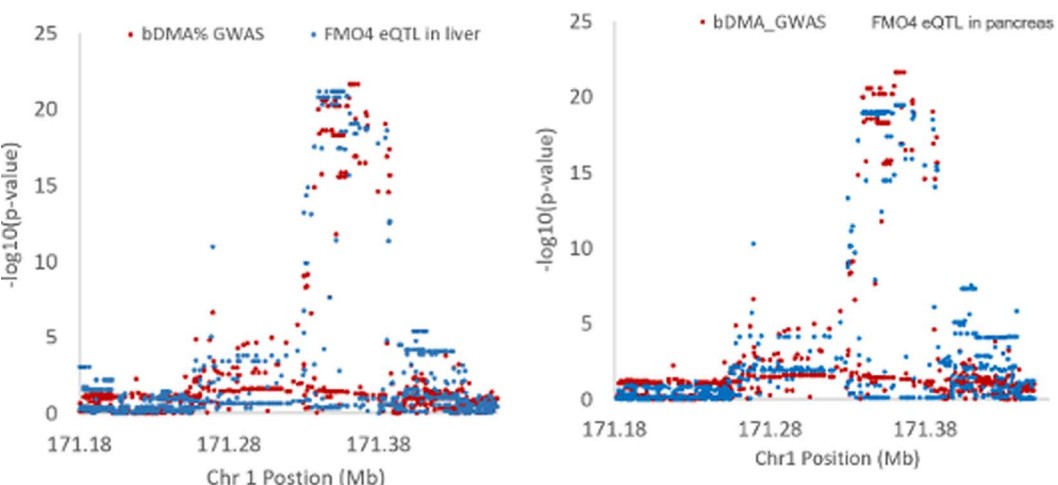

**Fig 7. Co-localization of the *FMO4* association signal for blood DMA% with a cis-eQTL for *FMO4* expression in GTEx A) pancreas tissue and B) liver tissue.**

## GWAS of arsenic-induced skin lesions

Using clinical data from genotyped participants from both HEALS and BEST (Bangladesh Vitamin E and Selenium Trial), we identified 3,448 participants with a diagnosis of arsenic-induced skin lesions (the most common sign of arsenic toxicity) and 5,207 participants without history of a diagnosis. We conducted a GWAS of arsenic-induced skin lesion status, and observed the strongest signal in the AS3MT region (P=6.9x10$^{-10}$; GC adjusted; P-value: 1.57x10$^{-8}$, S5 Fig). For the *AS3MT* and *FTCD* SNPs that impact arsenic species in both urine and blood, the DMA%-decreasing alleles showed consistent evidence of association with increased skin lesion risk (Table 1), as we have reported previously [19] However, for the newly identified SNPs in the FMO and GSTO gene clusters, clear evidence of association with skin lesion risk was not observed. Mediation analyses were conducted for the 3 skin lesion-associated SNPs (rs4919690, rs191177668, rs6173583). We found that adjustment for DMA% substantially attenuates the association between these SNPs and skin lesion status (S4 Fig). These results support the hypothesis that genetic variability in arsenic metabolism genes impacts arsenic-related toxicity risk, with AME acting as a mediator.

## Discussion

Using data from arsenic-exposed participants in Bangladesh, we performed GWAS of the relative abundances of three classes of arsenic species (iAs%, MMA%, and DMA%), measured in both urine and blood, and identified the FMO and GSTO gene clusters as novel regions in which genetic variation influences arsenic species composition in humans. While the variants we've identified previously (SNPs in the *AS3MT* and *FTCD* region) show association with arsenic species measured in both blood and urine, [45] the regions identified in this study show detectable association with arsenic species only in blood (*GSTO1, FMO4*) or only in urine (*FMO3*). Furthermore, the *AS3MT* and *FTCD* SNPs affecting arsenic metabolism show clear associations with risk of arsenic-induced skin lesions, while our newly identified metabolism-related variants (*GSTO1, FMO3,* and *FMO4*) do not.

Our observation that genetic effects on arsenic species can be detectable in blood, but apparently absent in urine (and vice versa), is somewhat unexpected given our prior findings that (1) *AS3MT* and *FTCD* SNPs have similar effects

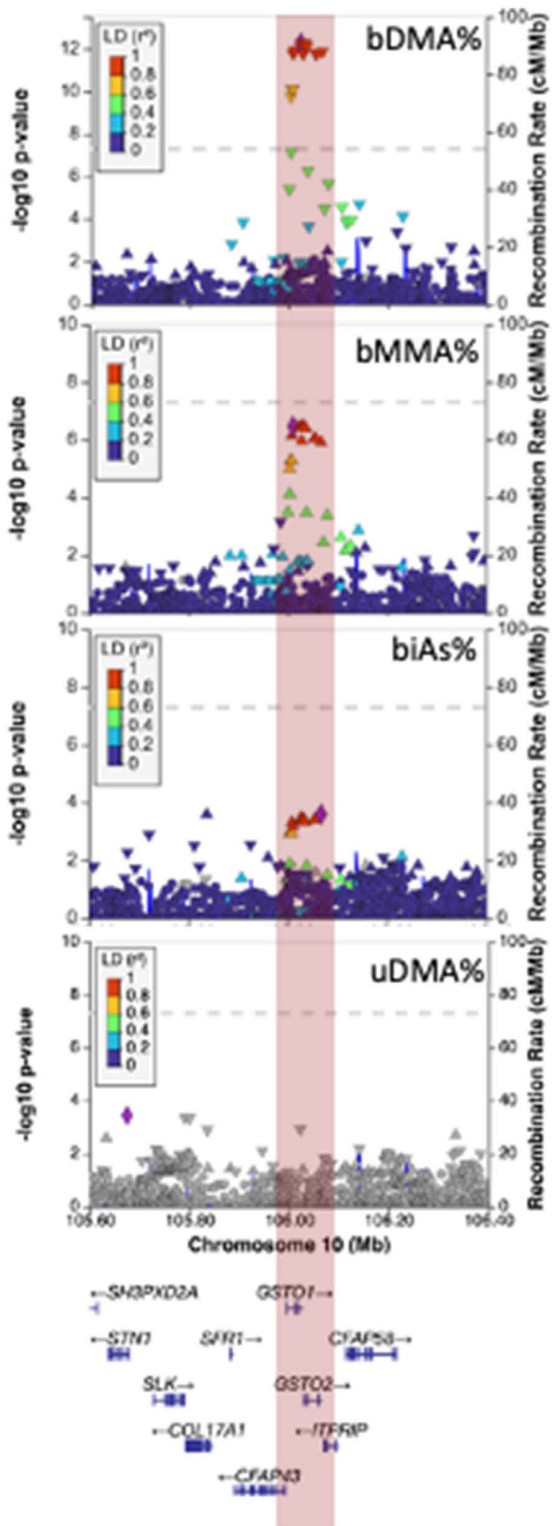

**Fig 8. Association of SNPs in the *GSTO1/GSTO2* region with arsenic species measured in blood, with no association observed for arsenic species measured in urine (uDMA%).**

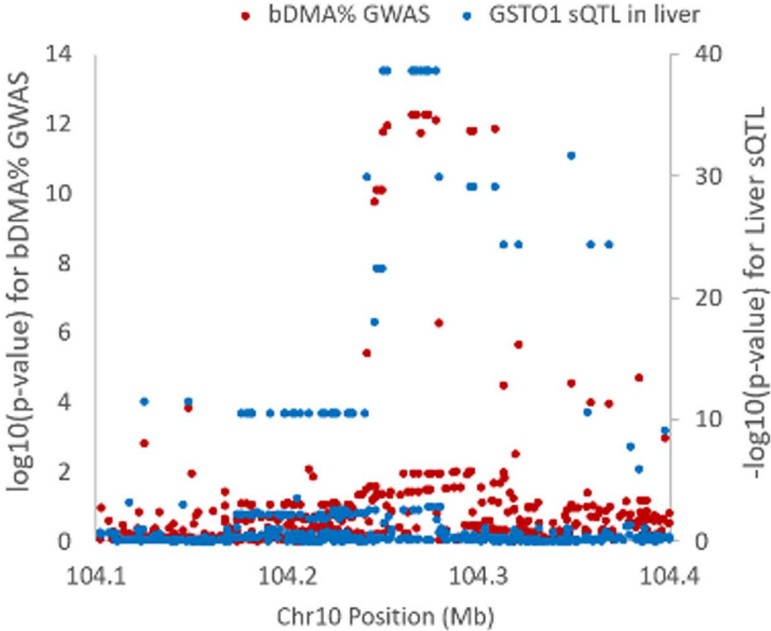

**Fig 9. Co-localization of *GSTO1* association signal for blood DMA% with a sQTL (chr10:104263077:104266084) for *GSTO1* in GTEx liver tissue.**

**Table 1. The association of arsenic metabolism efficiency (AME) SNPs and risk of arsenic-induced skin lesions\*.**

| | New Association Signals | | | Previously-identified association signals | | | |
|---|---|---|---|---|---|---|---|
| | *FMO3* rs12406572 | *FMO4* rs10912834 | *GSTO1* rs34521730 | *AS3MT* rs4919690 | *AS3MT* rs11191492 | *AS3MT* rs191177668 | *FTCD* rs61735836 |
| **Low efficiency allele (frequency)** | C (0.94) | C (0.44) | T (0.10) | C (0.13) | A (0.84) | T (0.01) | T (0.08) |
| **Tissue type(s) in which SNP affect AME** | Urine only | Blood only | Blood only | Urine and Blood | Urine and Blood | Urine and Blood | Urine and Blood |
| **Direction of effect on DMA%** | ↓ | ↓ | ↓ | ↓ | ↓ | ↓ | ↓ |
| **Skin Lesion Association** | | | | | | | |
| Odds Ratio | 1.1 | 1.06 | 0.88 | 1.38 | 0.94 | 2.32 | 1.36 |
| P-value | 0.17 | 0.12 | 0.04 | $6.9 \times 10^{-10}$ | 0.2 | $1.7 \times 10^{-5}$ | $4.1 \times 10^{-6}$ |

\* Skin lesion GWAS based on 3,448 skin lesion cases and 5,207 controls.

on metabolites in blood compared to urine and (2) arsenic species percentages (e.g., DMA%) measured in blood are positively correlated with the same metabolite percentage measured in urine [45]. These prior findings suggest that the arsenic species composition of urine reflects the composition of the blood that is being filtered by the kidney. However, our new findings point to complexities in arsenic metabolism, distribution, transport, and/or excretion that we do not yet fully understand. Such complexities could involve gene/enzyme functions that vary by cell or tissue type, important variability in the valence states of arsenic species that we are unable to measure in large numbers of human samples, and/or alternative pathways of metabolism and elimination (e.g., gut). Further research is needed to elucidate the precise mechanisms underlying these associations that differ between blood and urine arsenic species.

The association signal observed for urine species at *FMO3* is unique because the association is more pronounced for MMA% compared to DMA% (and undetectable for iAs%). For all other regions identified to date (*AS3MT, FTCD, FMO4,*

and *GSTO1*), the associations observed are most prominent for DMA%, suggesting a different toxicokinetic impact of the *FMO3* causal variant compared to the other regions.

FMOs are known to be involved in the oxygenation of xenobiotics, but they have not previously been reported to have a role in arsenic metabolism. Oxidation by FMOs could potentially play a role in reversing the *GSTO1*-catalyzed reduction of arsenic species, which according to the Challenger pathway, is required prior to arsenic methylation. However, alternative pathways of arsenic metabolism have been proposed [9,10] in which trivalent arsenic species are directly methylated, without oxidation. The resulting trivalent methylated species can then undergo oxidation to form pentavalent species, but pentavalent species would then need to be reduced to trivalent species to undergo methylation (under the alternative pathway). Thus, under either the Challenger or alternative pathways, FMOs could play a role in determining the ratio of trivalent to pentavalent species, thus impacting the supply of arsenic species that can be methylated. However, additional research is needed to understand the role of FMO enzymes in the metabolism of arsenic.

SNPs in FMO genes have been reported in GWAS to be associated with blood cell traits, [46] metabolomics phenotypes [47], and hormonal phenotypes [48]. For example, our top *FMO3* SNP (rs12406572) has been reported as a metabolite QTL for methylcysteine [46]. Mutations in *FMO3* cause trimethylaminuria, also known as "fish odor syndrome", a condition in which a defective *FMO3* enzyme causes accumulation of trimethylamine, which results in a distinctive odor resembling rotten fish [49].

*FMO2* has recently been shown to play a role in one-carbon metabolism in C.*elegans*, [50] an essential metabolic pathway that encompasses the folate and methionine cycles and provides one-carbon units for methylation reactions. This suggests the possibility that FMOs could influence arsenic methylation through effects on one carbon metabolism and the supply of methyl groups.

The associations identified for SNPs in the 10q25.1 region are likely to affect the function of *GSTO1*, a gene known to reduce pentavalent arsenic species to trivalent forms. Previous studies of candidate variants in *GSTO1* [51,52] have provided only modest evidence of their involvement in risk of arsenic toxicity. Additionally, a study of variation in four GST genes among three Ecuadorian populations suggested that two non-synonymous *GSTO1* variants may be under selective pressure, potentially due to environmental arsenic exposure [53]. However, the present study is the first to demonstrate a clear association between genetic variation in this region and arsenic species composition in human samples.

Previously identified SNPs in the *AS3MT* [54] and *FTCD* [19] regions show clear association of DMA%-decreasing alleles with elevated risk of arsenic-related skin lesions. This finding supports the hypothesis that genetic variants that decrease individuals' capacity to methylate arsenic, hence reducing the elimination and increasing the internal dose of arsenic, will increase the risk of arsenic-related health conditions. However, newly identified SNPs in the FMO and GSTO gene clusters did not show clear associations with arsenic-induced skin lesion risk. This lack of association could be due to power limitations, but it is also possible that SNPs influencing the reduction (or oxidation) of arsenic species, as opposed to methylation, may influence the distribution of pentavalent versus trivalent species in human tissues, with pentavalent species being more toxic, particularly MMA$^{III}$ [55]. While we cannot capture such effects in this study, they could have implications for toxicity risk, adding a layer of complexity to our underlying hypothesis that increasing DMA production should decrease toxicity risk.

For colocalization analyses, we leveraged eQTL and sQTL data from GTEx, a study of tissue donors largely of European ancestry. While the ancestry (and associated LD patterns) of GTEx are not well-matched to HEALS participants of Bangladeshi ancestry, our colocalization analyses produced strong posteriors, despite the LD mismatch.

Future directions for this research include replication of the reported associations in other populations with arsenic exposure and evaluating our metabolism-related SNPs for association with other arsenic-related health outcomes. Additional research is also needed to characterize the molecular mechanisms by which the identified variants impact gene function, their relevant cellular contexts, and their roles in pathways involved in arsenic metabolism. Importantly, identifying genetic variants associated with arsenic metabolism has the potential to inform precision public health strategies. By

pinpointing genetically susceptible populations, these findings could support targeted interventions, such as nutritional supplementation or enhanced exposure monitoring, aimed at reducing arsenic-related health risks [29,30].

## Supporting information

**S1 Fig. Distribution of As metabolites. A) Urine As Species** (Total n = 6,540), Columbia, n = 3, 687, NIAT, n = 163, FACT, n = 594, FOX, n = 341, Alberta, n = 1800**. B) Blood As Species (Total** n = 977), NIAT, n = 110, FOX, n = 273, FACT, 594.
(TIFF)

**S2 Fig. Conditional Analysis of the *FMO3* signal (n = 6,540).** A) uDMA% results adjusted for lead SNP rs12406572. B) uMMA% results adjusted for lead SNP rs12406572.
(TIFF)

**S3 Fig. Association of SNPs in the AS3MT region with A) bDMA%, bMMA%, and biAs% (n = 1,099); B) uDMA%, uMMA%, and uiAs% (n = 6,540).**
(TIFF)

**S4 Fig. Conditional Analysis of the *FMO4* region (n = 1,099).** A) bDMA% adjusted for lead SNP rs10912834. B) bMMA% adjusted for lead SNP rs10912834 B) biAs% adjusted for lead SNP rs2011345.
(TIFF)

**S5 Fig. Arsenic induced skin lesions GWAS.**
(TIFF)

**S1 Table. FMO3 sQTL Colocalization results.**
(XLSX)

**S2 Table. FMO4 eQTL Colocalization results.**
(XLSX)

**S3 Table. GSTO1 sQTL Colocalization results.**
(XLSX)

**S4 Table. Mediation Analysis demonstrating that associations between SNPs and arsenic-induced skin lesions are attenuated by adjustment for urine DMA% (representing AME).**
(XLSX)

**S1 Checklist. Inclusivity in global research questionnaire.**
(DOCX)

## Author contributions

**Conceptualization:** Lizeth I. Tamayo, Brandon L. Pierce.

**Data curation:** Tetiana Davydiuk, Donald Vander Griend, Syed Emdadul Haque, Tariqul Islam, Farzana Jasmine, Muhammad G. Kibriya, Joseph Graziano, X. Chris Le, Habibul Ahsan, Mary V. Gamble.

**Formal analysis:** Lizeth I. Tamayo, Lin Tong.

**Investigation:** Lizeth I. Tamayo.

**Methodology:** Syed Emdadul Haque, Tariqul Islam.

**Supervision:** Brandon L. Pierce.

**Visualization:** Lizeth I. Tamayo, Lin Tong.

**Writing – original draft:** Lizeth I. Tamayo.

**Writing – review & editing:** Lizeth I. Tamayo, Lin Chen, Habibul Ahsan, Mary V. Gamble, Brandon L. Pierce.

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
