## [Decision Letter · Decision Letter 0]

23 Dec 2024

PGENETICS-D-24-00757

Genetic variation in the FMO and GSTO gene clusters impacts arsenic metabolism in humans

PLOS Genetics

Dear Dr. Tamayo,

Thank you for submitting your manuscript to PLOS Genetics. The manuscript was fully evaluated at the editorial level and by independent peer reviewers. As you can see, the editors and reviewers appreciated the attention to an important problem, but also raised some concerns regarding the analyses and the interpretation of the current manuscript. Based on the reviews, we invite you to submit a revised manuscript that addresses these points. 

Please submit your revised manuscript within 30 days Jan 22 2025 11:59PM. If you will need more time than this to complete your revisions, please reply to this message or contact the journal office at plosgenetics@plos.org. Please include the following items when submitting your revised manuscript:

We look forward to receiving your revised manuscript.

Kind regards and happy holidays!

Weichun Huang

Guest Editor

PLOS Genetics

Hua Tang

Section Editor

PLOS Genetics

Aimée Dudley

Editor-in-Chief

PLOS Genetics

Anne Goriely

Editor-in-Chief

PLOS Genetics

**Additional Editor Comments:**

This study explored genetic variations in the FMO and GSTO gene clusters and their impact on arsenic species composition in humans, providing novel insights into arsenic metabolism. While the methodology was thorough and clearly presented, key areas for improvement remain to enhance the manuscript’s overall impact and clarity.

The reviewers acknowledged the study’s relevance, innovative approach, and robust methodology, but they emphasized the need to better articulate its broader motivation and implications. In particular, they recommended highlighting how these findings could inform practical interventions, such as tailoring strategies to reduce arsenic toxicity based on genetic profiles. Additionally, suggestions for refinement include clarifying key concepts, improving methodological transparency, and addressing design limitations.

Specific recommendations include:

1) Clarifying the role of GSTO1 in arsenic metabolism, ensuring accurate representation of its function.

2) Providing a detailed explanation of the detection methods for arsenic species to ensure methodological reliability.

3) Including distributions of arsenic metabolites across studies to contextualize variability in arsenic metabolism efficiency.

4) Conducting mediation analyses to better link genetic variants, arsenic metabolism, and health outcomes like skin lesions.

5) Addressing potential population substructure in the study design to enhance the robustness of results.

Incorporating these improvements will strengthen the manuscript and its contributions to understanding arsenic metabolism and its implications for human health.

**Journal Requirements:**

1) Please provide an Author Summary. This should appear in your manuscript between the Abstract (if applicable) and the Introduction, and should be 150-200 words long. The aim should be to make your findings accessible to a wide audience that includes both scientists and non-scientists. Sample summaries can be found on our website under Submission Guidelines:

https://journals.plos.org/plosgenetics/s/submission-guidelines#loc-parts-of-a-submission

4) We notice that your supplementary Figures are included in the manuscript file. Please remove them and upload them with the file type 'Supporting Information'. Please ensure that each Supporting Information file has a legend listed in the manuscript after the references list.

5) Please provide a complete Data Availability Statement in the submission form, ensuring you include all necessary access information or a reason for why you are unable to make your data freely accessible. If your research concerns only data provided within your submission, please write "All data are in the manuscript and/or supporting information files" as your Data Availability Statement.

6) Please ensure that the funders and grant numbers match between the Financial Disclosure field and the Funding Information tab in your submission form. Note that the funders must be provided in the same order in both places as well. State what role the funders took in the study. If the funders had no role in your study, please state: "The funders had no role in study design, data collection and analysis, decision to publish, or preparation of the manuscript.".

**Reviewers' comments:**

Reviewer's Responses to Questions

**Comments to the Authors:**

Reviewer #1: Review for “Genetic variation in the FMO and GSTO gene clusters impacts arsenic metabolism in humans”

Tamayo and colleagues performed GWA and colocalization analyses to test the association between arsenic species and common genetic variants. These analyses replicated previously observed relationships with arsenic and genetic variants in AS3MT and FTCD. In addition, novel variants were observed in FMO3, FMO4, and GSTO1. Colocalization analyses showed colocalization of these variants with several tissues in GTEx.

General Comments:

This study provides additional insight into potential mechanism of arsenic processing in the human body.

Methods and Materials:

As arsenic was measured at multiple timepoints, do you know if the concentration of arsenic is consistent across time points? Have you considered a longitudinal GWAS if not?

Results:

Statistical Analysis

Page 6: Are you able to conduct a sensitivity analysis utilizing the environmental factors mentioned in the introduction? (Smoking, BMI, …) Either by controlling for these factors or performing a subgroup analysis.

Is there concern for population substructure in these analyses? I don’t see the traditional PCA.

Have you considered a conditional GWAS to test the signals found?

Reviewer #2: Summary: This study identified the FMO and GSTO gene clusters as novel regions in which genetic variation influences arsenic species composition in humans. However, those newly identified metabolism-related variants did not show clear associations with arsenic-induced skin lesion risk.

Although the topic is interesting, there are several issues to be addressed to further improve the quality of the manuscript.

1. The statement in the conclusion “FMOs are involved in oxidation of xenobiotics, but have no known role in arsenic metabolism, while GSTO1 has a role in reducing arsenic species” is not very accurate. GSTO1 having a well-established role in catalyzing the reduction of arsenic species, but not in reducing arsenic species. The minor allele (T, MAF=10.3%) at lead SNP rs34521730 was associated decreased blood DMA%, increased blood MMA%, and increased blood iAs%, which means GSTO1 SNP is associated with reduced arsenic metabolism efficiency (AME).

2. In the methodology section, it is necessary to provide a detailed introduction to the detection methods of arsenic species in different studies, as this is the most important factor in ensuring the stability and reliability of research results.

3. The pooled distributions of DMA%, MMA%, and iAs% in urine and blood from different studies are shown in Supplementary Figure 1. Please provide the distribution of arsenic species in different studies, so that the readers can better understand the differences of arsenic metabolism efficiency among different studies and different bio-samples.

4. While the authors identified some associations between various SNPs and arsenic metabolism, and some of these SNPs are linked to arsenic-related skin lesions. Could the author include mediation analysis to further clarify the relationship between genetic variants, arsenic metabolism, and skin lesions?

Reviewer #3: This is a well-conducted, clearly written study that aims to identify genetic loci that influence arsenic metabolism. The novel aspect of the study is the use of both blood and urine arsenic metabolites. The study population is a highly relevant one with high exposures and arsenic-related skin lesions. The methods and results are clearly presented.

My main comment to the authors is that I am not clear what the motivation for this analysis is, or what we have now learned from the study that would get us close to an end goal, which presumably, is to tailor interventions for reducing arsenic toxicity to groups based on their genetic profiles. These issues could be addressed with some minor revisions to the introduction and discussion.

For example, in the last paragraph of the introduction, the authors state, "Given the significant global health impact of arsenic exposure and the variability in AME among individuals, our objective is to identify novel genetic determinants of arsenic metabolism and toxicity phenotypes." After reading the statement, my question was "Why?" -- how would this be a step toward an intervention. In the second paragraph of the discussion, "However, our new findings point to complexities in arsenic metabolism, distribution, transport and/or excretion that we do not yet fully understand." Yes, that's true, but why is that important? What implication could these have that might be important for the development of strategies that protect human health.

A very minor comment about terminology -- I am confused by the phrasing "GWAS of skin lesions," or "GWAS of urine species." It might be more precise to say a GWAS that focuses on detecting genetic variants that lead to differences in arsenic metabolites pattern in the urine (or something similar) at the first use, before going to the shorthand.

**Have all data underlying the figures and results presented in the manuscript been provided?**

Reviewer #1: Yes

Reviewer #2: Yes

Reviewer #3: Yes

PLOS authors have the option to publish the peer review history of their article (what does this mean? ). If published, this will include your full peer review and any attached files.

**Do you want your identity to be public for this peer review?** For information about this choice, including consent withdrawal, please see our Privacy Policy .

Reviewer #1: No

Reviewer #2: No

Reviewer #3: No

**Figure resubmission:**
---

## [Editor Report · Decision Letter 1]

31 Jul 2025

Dear Dr Tamayo,

We are pleased to inform you that your manuscript entitled "Genetic variation in the FMO and GSTO gene clusters impacts arsenic metabolism in humans" has been editorially accepted for publication in PLOS Genetics. Congratulations!

Yours sincerely,

Weichun Huang

Guest Editor

PLOS Genetics

Gregory Cooper

Section Editor

PLOS Genetics

Aimée Dudley

Editor-in-Chief

PLOS Genetics

Anne Goriely

Editor-in-Chief

PLOS Genetics

Comments from the reviewers (if applicable):

The authors have adequately addressed all concerns raised by reviewers, so I recommend the manuscript be accepted for publication.

**Data Deposition**

http://datadryad.org/submit?journalID=pgenetics&manu=PGENETICS-D-24-00757R1

**Press Queries**

---

## [Editor Report · Acceptance letter]

PGENETICS-D-24-00757R1

Genetic variation in the FMO and GSTO gene clusters impacts arsenic metabolism in humans

Dear Dr Tamayo,

We are pleased to inform you that your manuscript entitled "Genetic variation in the FMO and GSTO gene clusters impacts arsenic metabolism in humans" has been formally accepted for publication in PLOS Genetics! Your manuscript is now with our production department and you will be notified of the publication date in due course.

With kind regards,

Anita Estes

PLOS Genetics

On behalf of:
